# Research Progress on Bacterial Membrane Vesicles and Antibiotic Resistance

**DOI:** 10.3390/ijms231911553

**Published:** 2022-09-30

**Authors:** Xiaofei Liu, Jinyang Xiao, Shuming Wang, Jinxia Zhou, Jiale Qin, Zhibo Jia, Yanfeng Wang, Zhigang Wang, Yongmin Zhang, Huifang Hao

**Affiliations:** 1State Key Laboratory of Reproductive Regulation & Breeding of Grassland Livestock, School of Life Science, Inner Mongolia University, Hohhot 010020, China; 2Inner Mongolia University Research Center for Glycochemistry of Characteristic Medicinal Resources, Department of Chemistry and Chemical Engineering, Inner Mongolia University, Hohhot 010020, China

**Keywords:** bacterial membrane vesicles, BMVs, biogenesis, antibiotic resistance, conceptually new antibiotics, drug-delivery vehicles

## Abstract

As a result of antibiotic overuse, bacterial antibiotic resistance has become a severe threat to worldwide public health. The development of more effective antimicrobial therapies and alternative antibiotic strategies is urgently required. The role played by bacterial membrane vesicles (BMVs) in antibiotic resistance has become a current focus of research. BMVs are nanoparticles derived from the membrane components of Gram-negative and Gram-positive bacteria and contain diverse components originating from the cell envelope and cytoplasm. Antibiotic stress stimulates the secretion of BMVs. BMVs promote and mediate antibiotic resistance by multiple mechanisms. BMVs have been investigated as conceptually new antibiotics and drug-delivery vehicles. In this article, we outline the research related to BMVs and antibiotic resistance as a reference for the intentional use of BMVs to combat antibiotic resistance.

## 1. Introduction

Antibiotics are among the most important advancements in medicine. Since the discovery of antibiotics, they have saved countless lives and contributed to the development of a variety of health-related technologies [1]. However, antibiotic overuse and abuse, bacterial evolution under antibiotic stress, and a few new antimicrobials in the pipelines of the pharmaceutical industry have resulted in the emergence of multidrug-resistant (MDR) bacteria [2,3,4]. As researchers become more aware of the varying degrees of MDR, terms such as MDR, extensively drug-resistant (XDR), and pandrug-resistant (PDR) have been introduced to characterize the different resistance patterns identified in resistant bacteria. MDR was defined as acquired resistance to at least one agent in three or more antimicrobial categories. XDR was defined as insensitivity to all other classes of antimicrobials other than class 1–2 antimicrobials. PDR was defined as complete resistance to all agents in all antimicrobial categories [5]. Because there are few effective treatments available, treating infections brought on by MDR organisms can be extremely difficult. Antibiotic resistance has become a serious clinical problem. More than 70% of pathogens causing hospital-acquired infections have at least one resistance to currently commonly used anti-infective drugs. Furthermore, antimicrobial resistance is projected to cause a USD 100 trillion decline in the global gross domestic product (GDP) by 2050, making it a major global economic threat [6].

To prevent a post-antibiotic era, the World Health Organization (WHO) has released its first-ever list of “priority pathogens” that are resistant to antibiotics. According to WHO, the list was developed in an effort to direct and promote the research and development of novel antibiotics. The WHO list is divided into three categories based on how urgently new antibiotics are needed: critical, high, and medium priority [7].

In the context of WHO’s guidelines and the wide spread of antibiotic resistance, scientists have performed extensive research. Recently, the role of bacterial membrane vesicles (BMVs) in antimicrobial resistance attracted the interest of researchers. BMVs are nanoparticles derived from the membrane components of Gram-negative and Gram-positive bacteria. Antibiotic stress seems to stimulate BMVs synthesis and secretion, and BMVs have an important role in the acquisition of resistance and bacterial survival in the antibiotic environment. An improved understanding of the role of BMVs may lead to new strategies for controlling antibiotic resistance.

In this paper, we review the discovery, the biogenesis of BMVs, the effect of antibiotics on BMVs production, the mechanisms by which BMVs promote antibiotic resistance, and the application of BMVs in the treatment of antibacterial.

## 2. Discovery of BMVs

BMVs are nanoparticles derived from bacterial membrane components. They contain a wide range of bioactive compounds such as proteins, lipids, nucleic acids, metabolites, etc. [8,9,10,11,12]. Commonly, BMVs that originate from Gram-negative bacteria are called “outer membrane vesicles” (OMVs) and those that originate from Gram-positive bacteria or archaea are referred to as “membrane vesicles” (MVs) [9,13,14,15,16] (Figure 1).

OMVs in Gram-negative bacteria are spherical nanoparticles with sizes ranging from 10 to 300 nm. They bud from the outer membrane (OM) of the bacterial envelope [10,17,18], wrapping around periplasmic material and pinching off without breaking the cell membrane [11,19,20]. The discovery of OMVs dates back to the 1960s, when Knox et al. observed that *Escherichia coli* produced spherical “particles” surrounding the bacteria. Knox et al. proposed that these microscopic spherical structures were derived from the OM [17,21].

Subsequent studies found that OMVs can be produced by almost all Gram-negative bacteria [22,23,24,25,26,27,28]. Furthermore, OMVs are released during all stages of bacterial culture growth [29], although their numbers and compositions may differ slightly depending on the growth conditions [30]. OMV secretion is a conserved trait across Gram-negative bacteria, both pathogenic and non-pathogenic [31].

In addition to OMVs, Gram-negative bacteria can also release “Outer-inner membrane vesicles” (OIMVs). Previously, Kadurugamuwa and Beveridge proposed a model to explain the presence of certain cytoplasmic components in native and gentamicin-induced OMVs from *Pseudomonas aeruginosa*. In this model, the peptidoglycan (PG) layer can be weakened by autolysins such that the inner membrane protrudes into the periplasm. Cytoplasmic components enter these vesicles and are pinched off from the cell surface along with the surrounding outer membrane. Eventually, complex BMVs containing the outer membrane and inner membrane as well as cytoplasmic components such as DNA are formed [32]. In 2013, Pérez et al. first observed these complex BMVs composed of outer and inner membranes secreted by the Antarctic bacterium *Shewanella vesiculosa* M7^T^ and named them OIMVs [33]. This new type of BMVs contains not only the cell’s outer membrane but also its plasma membrane and cytoplasmic contents, thus possessing the ability to capture DNA. This finding confirms the model proposed by Kadurugamuwa and Beveridge. With the in-depth study of OIMVs, it was confirmed that OIMVs are also secreted by Gram-negative bacteria such as *Neisseria gonorrhoeae*, *P. aeruginosa* PAO1, and *Acinetobacter baumannii* AB41 [34]. This confirmation of OIMVs expands the so-far unified definition of BMVs in Gram-negative bacteria. However, due to convention and for simplicity, we still refer to BMVs released by Gram-negative bacteria as OMVs.

For decades, the idea of BMVs from Gram-positive bacteria was dismissed under the assumption that the strong, thick cell walls of Gram-positive bacteria would hinder the release of BMVs [35,36]. It was not until 2009 that Lee et al. demonstrated that *S. aureus* could release BMVs [37]. According to subsequent in-depth studies of other Gram-positive bacteria, BMVs production occurs in several species of Gram-positive bacteria, such as *Listeria monocytogenes*, *Enterococcus faecium*, *Mycobacterium ulcerans*, *Bacillus* spp., and *Streptococcus*, and *Lactobacillus* spp. are widely conserved [38,39,40,41,42,43,44,45,46,47]. Gram-positive BMVs are produced by protruding and selectively encapsulating various components from a section of their cytoplasmic membrane [48,49]; hence, it was named cytoplasmic membrane vesicles (CMVs). However, they are usually referred to as MVs [50].

## 3. Biogenesis of BMVs

BMV biogenesis is a physiological process. BMV biogenesis appears to follow main pathways: single cell-based vesicle release, which results in classical OMVs via OM blebbing, and cell lysis-mediated vesicle release, which results in the formation of OIMVs, explosive outer membrane vesicles (EOMVs), and MVs [51,52,53]. In addition, gene control also appears to be involved in MVs biogenesis, as shown in Figure 2.

### 3.1. Biogenesis of OMVs

The biogenesis of OMVs is now considered to be a fundamental, well-regulated, and perhaps conserved process [17]. Kulp and Kuehn proposed three possible models for OMVs biogenesis via the OM blebbing pathway (Figure 3) [17,54,55]. As the inner membrane remains intact, cytoplasmic components have no direct access to these OMVs [51]. In addition to the OMVs biogenesis caused by OM blebbing, explosive cell lysis can also trigger OMVs biogenesis (Figure 2) [51,53]. We will introduce them in Section 3.1.1 and Section 3.1.2 below, respectively.

#### 3.1.1. Trigger OM Curving

The localized elevation of several OM regions is an early step in the generation of OMVs and may even be the first step [17,56]. Kulp and Kuehn proposed three possible models for the biogenesis of OMVs via the OM blebbing pathway (Figure 3) [17,54,55]. The first model is the release of OMVs from the localized region where OM and PG layers lose their connection, as shown in Model A of Figure 3 [17,56]. This model is the primary mechanism for the generation of OMVs in “stress-free” environments, both as a type 0 secretion system (e.g., cytolysin A) and to support normal physiological processes such as membrane renewal.

The second model is the release of OMVs through outward pressures on the OM caused by the accumulation of periplasmic proteins or peptidoglycan fragments in the periplasmic space between OM and PG, as shown in model B of Figure 3 [57,58,59]. In a study of OMVs in *Porphyromonas gingivalis*, unbalanced PG turnovers in *P. gingivalis* periplasm led to an accumulation of cytosolic acids. Peptidoglycan and cytosolic acids then placed outward pressure on the OM, resulting in bulges in the OM and the release of OMVs [18,60].

The third model is that when curvature-inducing molecules aggerate, OM-PG connections may be broken or moved, resulting in the local bulging of OM, as in model C of Figure 3. Mashburn and Whiteley discovered variations in the curvature of the OM that contribute to the production of OMVs in *P. aeruginosa* [61]. Pseudomonas Quinolone Signal (PQS), a quorum-sensing molecule, is also implicated in the formation of OMVs. Secreted PQS interacts with the lipid A component of lipopolysaccharide (LPS) and accumulates negative charges to generate repulsion. This causes changes in membrane curvature, which are proportional to local PQS concentration, and ultimately leads to OMV formation [61,62,63,64]. Similar QS molecules have been found in the BMVs produced by *Vibrio shilonii*, *Paracoccus denitrificans*, etc. [65,66].

#### 3.1.2. Explosive Cell Lysis

Explosive cell lysis is triggered by phage-derived endolysin that degrades the peptidoglycan cell walls [51] or can be caused by other damage to the peptidoglycan component. Broken membrane fragments then aggregate and form EOMVs and OIMVs. In contrast to OMVs formed by blebbing, EOMVs randomly contain cytoplasmic components. Turnbull et al. revealed that *P. aeruginosa* explosive cell lysis forms vesicles by such vesiculation of shattered membrane fragments [67]. In addition, explosive cell lysis can generate OIMVs. Devos, S. et al. showed that treatment of multidrug-resistant *Stenotrophomonas maltophilia* with ciprofloxacin resulted in the release of bacteriophages and phage tail-like particles, cell lysis, and the formation of OIMVs that were enriched in cytosolic proteins [68].

### 3.2. Biogenesis of MVs

The biogenesis of MVs in Gram-positive bacteria differs from that of OMVs, and the process is still unclear. Heat-inactivated bacteria have shown to be unable to synthesize MVs when tested in MVs isolation experiments, implying that MV generation is dependent on metabolically active cells. The fluidity of the cell membrane and the integrity of the cell wall are key considerations in MVs release [69]. As shown in Figure 2, certain peptidoglycan hydrolases of Gram-positive bacteria can weaken the PG layer, leading to the formation of MVs across the cell wall [70]. The expression of an endolysin encoded by a defective prophage trigger in *Bacillus subtilis* triggers pores in the peptidoglycan’s cell wall that allow MVs to escape [71]. Gene control is also important for MVs generation [72], as evidenced by the Gram-positive human pathogen *Streptococcus pyogenes*, also known as group A streptococcus (GAS). It was discovered to be negatively regulated by the CovRS two-component system [46]. The Pst/SenX3-RegX3 signaling pathway has also been found to regulate the formation of MVs in *Mycobacterium tuberculosis* [37,73]. Rather than relying on a small collection of genes, the synthesis of MVs relies on a complex network of genes [69].

In addition, researchers have found that the biogenesis of BMVs is closely related to several stressful conditions. Previous research has demonstrated that BMV secretion levels were affected by physiological or environmental stressors such as oxidative stress, high temperature, and antibiotic treatment and that stressors can lead to changes in BMV compositions [17,32,74,75].

## 4. Secretion of BMVs in Response to Antibiotic Stress

As discussed previously, the secretion and composition of BMVs are affected by and can be manipulated through physiological and environmental stressors, such as antibiotic treatment. The treatment of bacteria with sublethal concentrations of certain antibiotics is a recognized trigger for the formation of BMVs. Antibiotic stress has been shown to enhance BMVs secretion in studies [76,77]. As described in Figure 4, three pathways by which antibiotic stress stimulates the formation of BMVs have been identified [51].

### 4.1. Bacterial Envelope Stress Caused by Antibiotics

Antibiotic treatment can cause an increase in pressure on the bacterial envelope. Antibiotics that cause this pressure, such as gentamicin and polymyxin, promote the formation of OMVs by triggering OM blebbing, as shown in Figure 4A [51,78]. Gentamicin is an aminoglycoside antibiotic that is supposed to kill bacteria by inhibiting protein synthesis. However, this cationic antibiotic can also perturb the packing order of lipids, which can lead to bilayered membrane instability. Gentamicin increases the release of OMVs by 3 to 5-fold in *P. aeruginosa* [79]. Polymyxins exert detergent-like activity on the cell wall. Bauwens et al. found that the antibiotics phosphomycin, meropenem, and polymyxin B increased the production of Enterohemorrhagic *E. coli* (EHEC) OMVs by acting as a source of bacterial envelope stress [74]. This result is consistent with previous reports that OMVs are induced by meropenem in *P. aeruginosa* [80] and by imipenem in *Stenotrophomonas maltophilia* [80,81].

### 4.2. Induction of SOS Response

The SOS response, which is an inducible DNA repair mechanism, is an important protective mechanism for bacteria that is triggered by DNA damage [82,83,84]. Antibiotic therapy sometimes exacerbates bacterial infections by inducing SOS response and increasing BMVs secretion. Some DNA-damaging antibiotics, especially quinolones, such as ciprofloxacin, can induce SOS response. The SOS response can in turn trigger the expression of endolysins encoded by prophages, resulting in lysis-stimulated vesicles formation, as shown in Figure 4B [51].

The mechanisms of bacterial-induced SOS response induced by *E. coli* and *P. aeruginosa* are described below. During the normal growth of bacteria, the LexA deterrent protein suppresses SOS gene expression. However, when DNA damage is significant, replication pauses, and the amount of single-stranded damaged regions in DNA increases. RecA binds to single-stranded DNA (ssDNA) induced by DNA-damaging agents and mediates the autocatalytic cleavage and the inactivation of the LexA blocker protein. Then the inhibition is derepressed, the SOS gene is triggered, and the SOS response is activated [83,84]. Thus, antibiotic therapy sometimes exacerbates bacterial infections by inducing SOS response and increasing BMV secretion.

The formation of BMVs has been linked to SOS response by several studies. Maredia et al. used ciprofloxacin to treat both wild-type and LexA non-cleavable (LexAN) *P. aeruginosa* strains (without induction of SOS response). Under ciprofloxacin treatments, wild-type strains secreted considerably more OMVs than LexAN strains. This implies that the antibiotic-induced SOS response is involved in the generation of BMVs [84]. Ciprofloxacin and mitomycin C as SOS response inducers greatly increased OMV production in EHEC O104:H4 and O157:H7 and delivered Shiga toxin 2a (Stx 2a) through OMVs. Furthermore, the synthesis of Stx 2a is associated with the induction of Stx prophages carrying toxin genes. Therefore, antibiotics that induce SoS response not only induce Stx production but also trigger explosive cell lysis that disperses toxins through OMVs [74]. Andreoni et al. showed that mitomycin C induced an SOS response in Gram-positive bacteria, triggering the formation of MVs in lysogenic *S. aureus* strains but not in phage-devoid counterparts [76]. All these findings suggest that the antibiotic-induced SOS response plays an important role in the biogenesis of BMVs [84].

### 4.3. Inhibition of Bacterial Cell Wall Biosynthesis

β-lactam antibiotics stimulate the formation of BMVs by weakening and generating pores in the PG layer, as shown in Figure 4C [85,86]. After treatment with β-lactam antibiotics, the cytoplasmic membrane and content protruded into the extracellular space and released MVs [71,87,88]. For example, Andreoni et al. exposed *S. aureus* strains to 10 times the minimum inhibitory concentration (MIC) of the β-lactam antibiotics flucloxacillin and ceftazidime and found a significant increase in the secretion of MVs [76].

## 5. BMVs-Mediated Antibiotic Resistance

Antibiotic resistance has become an increasingly serious concern as a result of antibiotic abuse. BMVs are implicated in virulence, pathogenicity, cell–cell communication, biofilm formation, and antibiotic resistance, among other bacterial biological processes [8,50,89,90]. Studies have shown that Peptidylarginine deiminases (PADs) inhibitors can be used to effectively reduce BMVs release, both in Gram-negative and Gram-positive bacteria. Importantly, this resulted in enhanced antibiotic sensitivities of both *E. coli* and *S. aureus* to a range of antibiotics tested. This implies that BMVs play a significant role in antibiotic resistance [91].

As illustrated in Figure 5 [50,90,92], BMVs can mediate antibiotic resistance in a variety of ways. First, BMVs can act as decoys to bind antibiotic compounds or isolate antibiotics. Second, BMVs can enhance drug effluxes and translocate antibiotics out of the cell. Third, BMVs are also encapsulated with antibiotic-degrading enzymes that can hydrolyze or chelate antibiotics extracellularly. Finally, BMVs can carry resistance genes that promote the spread of drug resistance by transferring and disseminating these resistance genes.

### 5.1. As “Decoys” to Bind Antibiotics or “Barriers” to Isolate Antibiotics

Bacterial membranes are the major component of BMVs. BMVs bind and uptake antibiotics and toxins because of the affinity of these compounds for bacterial membranes, as illustrated in Figure 5A [19,92]. OMVs released by *E. coli* and *P. aeruginosa* can act as decoy receptors for antimicrobial drugs such as colistin and polymyxin B, allowing bacteria to survive would-be lethal doses [29,87,93].

*S. aureus* secretes MVs enriched with penicillin-binding proteins that ordinarily bind to β-lactam drugs and cause methicillin resistance in Gram-positive bacteria [37]. In vitro and in whole blood, Andreoni discovered that isolated MVs of *S. aureus* protected the bacteria against death by the membrane-targeted antibiotic daptomycin. Thus, antibiotic-induced MVs are considered to act as decoys, helping bacteria to survive [76].

Furthermore, OMVs can also function as barriers to provide interbacterial glue to form nearly impenetrable multicellular structures, such as biofilms, which confers resistance to antibiotics and other antibacterials [56]. *P. aeruginosa* OMVs with β-lactamase activity produce a biofilm in the lungs of cystic fibrosis patients, shielding the bacteria underneath from medications [94]. OMVs purified from *E. coli* MG1655 protected *P. aeruginosa* and *A. baumannii* from death by the membrane-active antibiotic colistin and melittin but not from other antibiotics with different mechanisms of action, such as ciprofloxacin, streptomycin, and trimethoprim [18,29,87]. Colistin was isolated by the OMVs, and melittin was degraded by the OMVs’ protein components. This implies two different mechanisms for resistance to membrane-active antimicrobials [87]. The OMVs of Antarctic bacterium *Pseudomonas syringae* Lz4W provided similar protections by scavenging the membrane-active antibiotics colistin and melittin, rendering their growth inhibition ineffective against the strain [95].

### 5.2. Transport of Antibiotics to the Outside of the Cell

Many bacteria can expel antimicrobial medications out of the intracellular compartment, leaving the intracellular drug concentration insufficient for antimicrobial action and resulting in drug resistance [96,97,98]. As shown in Figure 5B, OMVs appear to promote the short-lived survival of susceptible bacteria in the antibiotic environment by removing the antibiotic from the bacterial cell and not allowing it to accumulate in sufficient concentrations required for an inhibitory effect. Vesicles produced by a ciprofloxacin-resistant mutant of the mycoplasma *Acholeplasma laidlawii* have been found to contain ciprofloxacin [96]. McBroom et al. have shown that high temperatures stimulate the formation of OMVs in *E. coli*, resulting in the removal of misfolded proteins induced by heat stress through packages moved into OMVs [57]. This is another mechanism by which OMVs transport harmful substances outside the cell to protect the parental bacterium.

### 5.3. Enzymes Carried by BMVs

BMVs produced by Gram-negative and Gram-positive bacteria can carry enzymes that degrade antibiotics, leading to antibiotic resistance [99,100], as shown in Figure 5C. Several studies have shown that BMVs carrying β-lactamases can provide temporary antibiotic resistance by degrading β-lactam antibiotics [8,29,99]. According to Kim et al., OMVs from β-lactam-resistant *E. coli* directly degraded β-lactam antibiotics, protecting susceptible *E. coli* strains from β-lactam antibiotic-induced growth suppression [101].

Protective effects can extend to the surrounding microbial community, such as when *Salmonella* spp. (Sal26B) and *Edwardsiella tarda* (ED45) conferred transient antibiotic resistance to neighboring species [102,103]. In the human respiratory tract, BMVs containing β-lactamase are protective not only of parental bacteria but also of some other coexisting bacteria [100,104]. OMVs carrying β-lactamases produced by *Bacteroides* spp., which constitute the majority of the human colonic microbiota, can protect commensal bacteria and enteric pathogens (such as *Salmonella typhimurium*) from β-lactam antibiotics [103].

Gram-positive bacteria use the same strategies. Blaz, a β-lactamase protein, can be released by *S. aureus* via MVs. Ampicillin-sensitive Gram-negative and Gram-positive bacteria can survive in the presence of ampicillin thanks to these MVs [105].

### 5.4. Drug-Resistance Genes Carried by BMVs

BMVs promote the development of long-term adaptive resistance to antibiotics through the horizontal gene transfer (HGT) of resistance genes, as depicted in Figure 5D [8]. OMVs isolated from the food-borne pathogen *E. coli* O157:H7 helped transfer genes to the recipient *S.enterica* serovar Enteritidis, imparting cytotoxicity and antibiotic resistance to recipient cells [106,107]. Carbapenem-resistant *A. baumannii* strains can secrete the plasmid-borne bla_OXA-24_ gene via OMVs and protect susceptible *A. baumannii* strains from the toxicity of antibiotics [108]. Fulsunder et al. noted the movement of DNA from the cytoplasm of the donor bacteria *Acinetobacter baylyi* JV26 to the periplasm, then into the OMV, and finally into the recipient bacteria *E. coli* DH5 and *A. baylyi* JV26 [109]. Recent research has shown that avian pathogenic *E. coli* (APEC) OMVs that produce CTX-M-55-type extended-spectrum β-lactamase (ESBL) can mediate the horizontal transfer of the bla_CTX-M-55_ gene [48]. The essential role of OMVs in HGT suggests that OMVs can deliver their DNA cargo into the bacterial cytoplasmic matrix [110]. However, it is unknown whether the OMV-mediated horizontal transfer of antibiotic resistance genes is prevalent.

Despite the presence of DNA in Gram-positive bacteria MVs, little is known about their role in HGT. Klieve et al. found that MV-mediated HGT was able to restore the ability of *Ruminococcus* spp. strain YE71 mutant to degrade crystalline cellulose. And this property was stable and heritable in subsequent bacterial generations. However, YE71 vesicles were unable to transform the hemicellulolytic ruminal bacterium *B. fibrisolvens* AR5. This suggests a possible mechanism for species-selective transformation [111]. It was later found that MVs could also facilitate gene transfer by mediating phage infection. In *Bacillus subtilis*, phage-resistant cells can gain phage sensitivity by acquiring phage receptors carried by MVs produced by susceptible bacteria [112]. However, further studies in other Gram-positive species are needed to confirm these findings.

## 6. Prospects and Directions for the Application of BMVs in Antibacterial

New drugs and strategies for fighting bacterial infection are urgently needed in the face of the widespread prevalence of antibiotic resistance. The therapeutic potential of BMVs in the treatment of bacterial antibiotic resistance has generated interest in recent years. BMVs are strain-secreted products that cannot grow or reproduce, and they can contain exogenous materials, so they can be developed as antibacterial therapeutic tools [113]. Recent breakthroughs in the use of BMVs in antibacterial applications are discussed here.

### 6.1. BMVs as Conceptually New Antibiotics

Kadurugamuwa and Beveridge found that both native (n-OMVs) and gentamycin-induced OMVs (g-OMVs) from *P. aeruginosa* contained a periplasmic 26-kDa autolysin (peptidoglycan hydrolase). Autolysins are a class of endogenous enzymes that hydrolyze different peptidoglycan linkages, impairing the integrity of the murein sacculus or cell wall in the process [114,115]. n-OMVs were able to kill gentamicin-resistant *P. aeruginosa* cultures. This suggests that the fusion of n-OMVs with the OM releases autolysins into the periplasm, where they degrade peptidoglycan and lyse cells. g-OMVs were more effective at lysing these resistant bacteria than n-OMVs or free antibiotics. Because they contain low levels of gentamicin in addition to autolysins, and they release gentamicin and autolysin into these resistant cells. In the context of increasing antibiotic resistance, these “predatory” OMVs could have other profound effects. They may be bacteriolytically active against both Gram-positive and Gram-negative pathogens [32]. The findings could help develop a conceptually new group of antibiotics designed to be effective against hard-to-kill bacteria [116].

### 6.2. BMVs as Antibiotic Drug Delivery Vehicles

Due to their satisfactory drug loading, and targeting ability, BMVs may be suitable as functional carriers for antibiotic drug delivery [117,118]. BMVs have at least the following advantages as natural drug delivery vehicles: (1) they fuse with target cells, particularly Gram-negative bacteria cell membranes; (2) they can be used for targeted drug delivery because they are naturally loaded biomolecules; (3) they are readily generated from parental bacteria and circulate in the blood for a long period of time [119,120]. Using BMVs for cell-specific medication delivery appears to be a viable option. The part of BMVs as drug delivery vehicles will be discussed in two parts: the delivery of endogenous encapsulation antibiotics and the delivery of exogenously added antibiotics.

#### 6.2.1. Delivery of Endogenous Encapsulation Antibiotics

One advantage of BMVs as delivery vehicles for endogenous encapsulation antibiotics is the delivery of molecules to the cytoplasm of target bacteria by bypassing the outer membrane barrier of antibiotic uptake by Gram-negative bacteria. OMVs have now been shown to deliver cargo across the Gram-negative cell envelope. This indicates that medication encapsulation within OMVs may lessen transport issues that limit the efficacy of many antibiotics against Gram-negative bacteria [110]. Moreover, the ability of OMVs to remove unwanted chemicals from cells has been used to load antibiotics into OMVs. This loading method is the incorporation of drugs into OMVs during their biological generation through parental bacteria. Antibiotic-containing OMVs can be created based on this [110,121].

Huang et al. identified an interesting antibiotic efflux mechanism mediated by multidrug-resistant *A. baumannii* OMVs and then designed novel antibiotic-loaded OMVs using this mechanism to protect against intestinal bacterial infection. They induced *A. baumannii* OMVs containing levofloxacin with sub-MIC concentrations of levofloxacin. Under this treatment, antibiotic stress resulted in the encapsulation of numerous intracellular components into OMVs by highly expressing efflux pumps in the bacteria. They discovered that OMVs containing levofloxacin were effective in killing enterotoxin-producing *E. coli* (ETEC), *K. pneumoniae*, and *P. aeruginosa*. In a mouse model of ETEC infection, they administered low-dose oral antibiotic-loaded OMVs to mice. They then found that OMVs containing levofloxacin were more effective than the free drug. Furthermore, antibiotics contained in OMVs do not travel fast to other uninfected organs, causing harm. Instead, the drugs accumulate in the small intestine, where they can be effectively delivered at the site of intestinal infection [122].

#### 6.2.2. Delivery of Exogenously Added Antibiotics

The current successful development of several active incorporation techniques, such as electroporation and sonication, enhanced the incorporation of drugs and other therapeutics into BMVs. Although these techniques have not been specifically used to load antibiotics into BMVs, they also demonstrate the potential of BMVs as antimicrobials through this approach [110].

## 7. Summary and Outlook

In this paper, we highlighted five important topics related to BMVs: their discovery, biogenesis, their response to antibiotics, mechanisms mediating antibiotic resistance, and their application as antimicrobials.

BMVs serve as a survival mechanism for bacteria in the face of adversity, such as antibiotic exposure. Antibiotic stress stimulates the production of BMVs. BMVs can then mediate antibiotic resistance by multiple mechanisms. BMVs hold promise for combating antibiotic resistance and treating MDR bacterial infections. Through genetic engineering and membrane modification, BMVs have been developed as drug delivery vehicles. In addition, studies have demonstrated that BMVs can serve as conceptually new antibiotics.

However, many aspects of the biology and potential utility of BMVs remain unclear. Although many studies have shown that BMVs can protect bacteria from antibiotics, it is unclear whether this protection is specific. The potential of BMVs as antibacterial therapeutic agents is still being explored. More in-depth investigations of BMVs and their relationship with antibiotics will certainly help address the challenge of antibiotic resistance in the future.

## Figures and Tables

**Figure 1 ijms-23-11553-f001:**
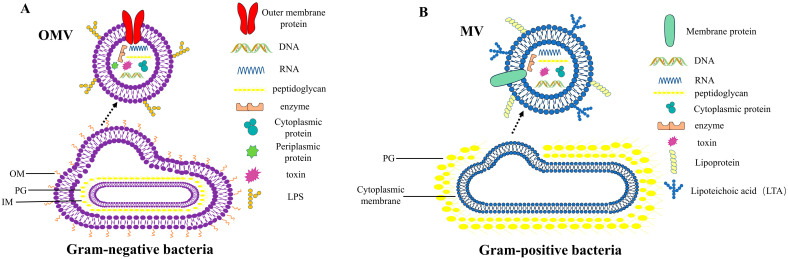
Structure and composition of BMVs. (**A**) OMVs and (**B**) MVs.

**Figure 2 ijms-23-11553-f002:**
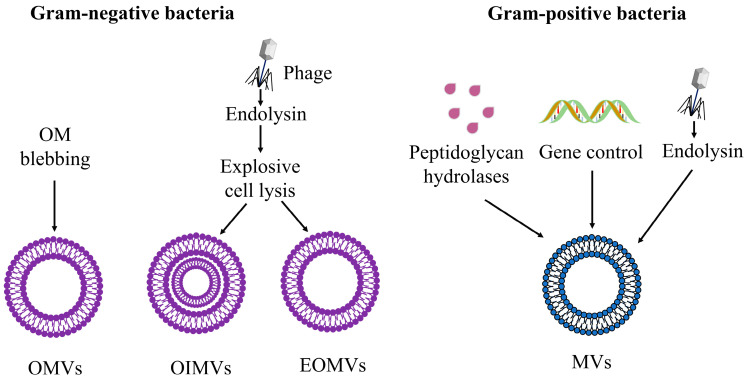
Different pathways for the biogenesis of BMVs.

**Figure 3 ijms-23-11553-f003:**
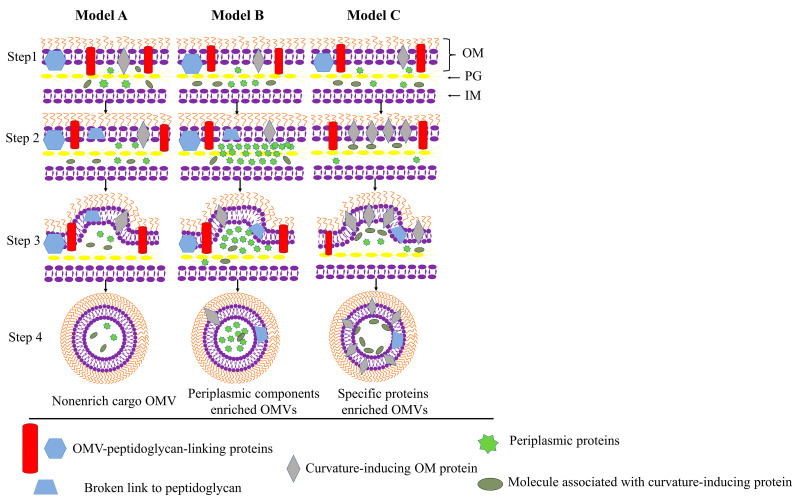
Biogenesis of OMVs via OM blebbing. Step 1: Gram-negative bacterial-cell-enveloped proteins are initially uniformly distributed. The outer membrane is linked with peptidoglycan. Steps 2 and 3: Vesiculation is initiated when the connection between the outer membrane and the peptidoglycan is lost due to the migration of connecting proteins or direct breakage. Models A, B, and C demonstrate three possible mechanisms of OMVs generation. Model A depicts OMV production at its most basic level. In model B, additional budding events can be generated by periplasmic protein aggregation. The resulting OMVs are enriched with periplasm cargo. In model C, the accumulation of curvature-inducing OM proteins causes OMVs to bud from the Gram-negative bacterium at specific proteins on the envelope surface. The OMVs will be enriched with curvature-inducing molecules and molecules associated with them (adapted from [55]). This figure is adapted with copyright permission.

**Figure 4 ijms-23-11553-f004:**
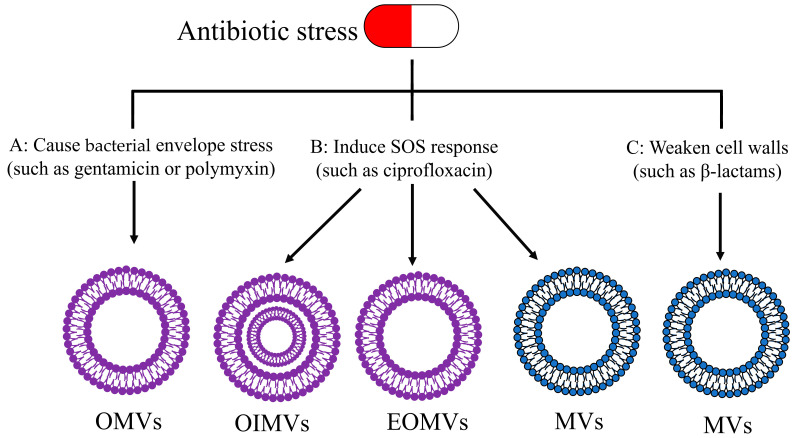
Three mechanisms of BMV formation under antibiotic stress.

**Figure 5 ijms-23-11553-f005:**
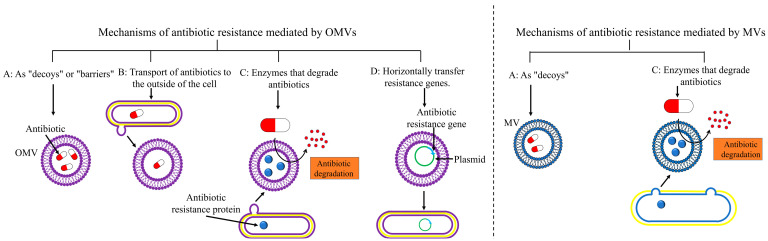
Mechanisms of antibiotic resistance mediated by BMVs. The left picture shows OMVs of Gram-negative bacteria, and the right picture shows MVs of Gram-positive bacteria.

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
