# Peer review of "Research Progress on Bacterial Membrane Vesicles and Antibiotic Resistance"

_ijms, 2022, doi:10.3390/ijms231911553_

Round 1

Reviewer 1 Report (Previous Reviewer 1)

My comments are enclosed within this mail.

Author Response

Reviewer 2 Report (New Reviewer)

In the review titled  “Research progress on Bacterial Membrane Vesicles and Antibiotics Resistance,” the authors have discussed the BMVs in the context of their discovery, biogenesis, response to antibiotics, mechanisms mediating antibiotic resistance, and their application as antimicrobials.

The paper delivers a piece of very important information in the field of AMR and infectious diseases.

I would like to add a few comments:

1- Overall language of the paper is not up to the mark, extremely long sentences have been used in the entire manuscript, and the narration is also not on point. This issue is diluting the quality of this impressive paper.

2- In the introduction section I didn't find a proper flow to connect the facts about AMR to BMVs.

Suddenly from the facts about AMR, and WHO's guidelines, the authors jumped to BMVs (Line 54).  I would suggest crafting a proper linkage so that readers could correlate.

3- The quality of the figures is not up to the mark. Figures 1, 3, and 5 are very small in size.

Author Response

Reviewer 3 Report (New Reviewer)

This paper reviewed the discovery, biogenesis of BMV, their response to antibiotics, mechanisms mediating antibiotic resistance, and their application as antimicrobials. The article is well written, well organized, and provides a balanced view of the topic.

I have minorscomments regarding the paper.

Most of the figures in this paper are adapted from the other published papers. As the paper topic is BMVs and antibiotics resistance, I suggest the authors provides a new schematic figure to show the interactions between BMVs and antibiotics resistance.

Round 2

Reviewer 1 Report (Previous Reviewer 1)

I thank the authors for their enthusiasm and the demonstrated will to undergo 3 revisions. Now all is clear and I have no further comments. 

This manuscript is a resubmission of an earlier submission. The following is a list of the peer review reports and author responses from that submission.

Round 1

Reviewer 1 Report

The review entitled “The Roles of Bacterial Membrane Vesicles as Resistance Acquisition Tools and the Fight against Antimicrobial Resistance” aims to present the versatile structural and functional roles of the bacterial membrane vesicles (BMVs) linked to antibiotic resistance.

My general impression is that the review in its current version, and for various reasons, fails to clearly set these objectives. More than 60% of the review covers generalities about BMVs and only 30% of it addresses the claims introduced by its title. Either the authors should change the title of their review or they should make the review more focused on BMV-mediated antibiotic resistance. Also, the authors have barely touched upon “The Fight against Antimicrobial Resistance” and I think they could have presented it in a much better light. Finally, most of the findings linked to BMV-mediated antimicrobial resistance are scattered throughout the review, diluting the main message.

Reviewer 2 Report

The title of the article refers to the role of vesicles in resistance to antibiotics and how they might be employed to fight antibiotic resistance. This topic occupies 3.5 of 12 pages of text and one of 6 figures.    The rest of the article goes into a lot of detail on biogenesis of vesicles, factors that promote vesicle formation in vitro and their cargo. Much of this has been covered by previous reviews. While being a mostly reasonable account this is too long. More should be written about the subject of the review in a critical fashion. 

Some of the sections are confusing because they describe both OMVs and MVs even in the same paragraph

Vaccines

On the question of the use of vesicles in vaccination the authors have missed an  important development in this area, namely the use of outer membrane vesicles of Neisseria meningitidis B in the widely used Bexsero vaccine against meningococcal disease. In this case any potential toxicity of LPS in the vesicle is clearly not a problem.

There is a mistake in the section of vaccines -  ref 136 does not refer to use of Burkholderia OMVs as a vaccine, rather they carry components with antibacterial activity against S.aureus 

More should be written on the potential use of bioengineered vesicles as vaccines for Gram  positive bacterial infections

Delivery of bioactive molecules

The section on BMVs as drug delivery vehicles should be separated into delivery of endogenously synthesized antibiotics  and those loaded with exogenously added  antibiotics. One advantage of the former might be to deliver molecules to the cytoplasm of the target bacteria by circumventing outer membrane barriers to antibiotic uptake in Gram negative bacteria. This and the transfer of plasmid DNA implies that vesicles can fuse with the membrane of intact bacteria and deliver cargo into the cytoplasm. This must be discussed

There should be more information about “drug loading” into vesicles.   Does this occur during vesicle formation or can vesicles taken up antibiotics ? Can drugs be loaded into MVs as well as OMVs? The relevance of liposomes (ref 129 line 436)  should be explained

Is there any evidence that  vesicles loaded with antibiotics can be used to treat infected animals?  In other words can vesicles be used in vivo?

There is  mention of BMVs, (both OMVs and MVs? ) being able to promote horizontal transfer of plasmids encoding antibiotic resistance.  This implies that the DNA containing vesicle can fuse with the (membrane of) a cell of another bacterium and deliver the plasmid DNA although this is not explicitly stated.

Round 2

Reviewer 1 Report

Please find my comments enclosed within this message.

Reviewer 2 Report

The authors have made extensive changes to the manuscript according to my comments and recommendations.  It is substantially improved